# OA-HybridCNN (OHC): An advanced deep learning fusion model for enhanced diagnostic accuracy in knee osteoarthritis imaging

Yihan Liao[1☉], Guang Yang[2☉], Wenjin Pan[1], Yun Lu[3,4]*

1 Department of anesthesiology, Xuzhou Medical University, Xuzhou, China, 2 Department of Neurology, Kunshan Hospital of Traditional Chinese Medicine, Kunshan, China, 3 Department of Min's Wound, Kunshan Hospital of Traditional Chinese Medicine, Kunshan, China, 4 School of Nursing, Yangzhou University, Yangzhou, China

☉YL and GY contributed equally to this work.
* ksluyun@126.com

## Abstract

Knee osteoarthritis (KOA) is a leading cause of disability globally. Early and accurate diagnosis is paramount in preventing its progression and improving patients' quality of life. However, the inconsistency in radiologists' expertise and the onset of visual fatigue during prolonged image analysis often compromise diagnostic accuracy, highlighting the need for automated diagnostic solutions. In this study, we present an advanced deep learning model, OA-HybridCNN (OHC), which integrates ResNet and DenseNet architectures. This integration effectively addresses the gradient vanishing issue in DenseNet and augments prediction accuracy. To evaluate its performance, we conducted a thorough comparison with other deep learning models using five-fold cross-validation and external tests. The OHC model outperformed its counterparts across all performance metrics. In external testing, OHC exhibited an accuracy of 91.77%, precision of 92.34%, and recall of 91.36%. During the five-fold cross-validation, its average AUC and ACC were 86.34% and 87.42%, respectively. Deep learning, particularly exemplified by the OHC model, has greatly improved the efficiency and accuracy of KOA imaging diagnosis. The adoption of such technologies not only alleviates the burden on radiologists but also significantly enhances diagnostic precision.

## 1. Introduction

Osteoarthritis (OA) is a degenerative joint disorder presenting with cartilage damage, subchondral bone remodeling, osteophyte formation, alterations in the joint capsule, and synovial inflammation [1–3]. With the aging population growing and obesity becoming increasingly prevalent globally, the incidence of OA has steadily risen. In 2024, approximately 7.6% of the global population were affected by OA. This is

**Data availability statement:** The anonymized dataset necessary to replicate the study findings has been uploaded as Supporting Information.

**Funding:** Jiangsu Key Laboratory of Human and Animal Diseases, "Protecting Xin" Fund Project (HX2402), Kunshan City Science and Technology Special (Social Development) Guiding Project (KSZ2338). The funders had no role in study design, data collection and analysis, decision to publish, or preparation of the manuscript.

**Competing interests:** The authors have declared that no competing interests exist.

projected to increase by 60–100% by 2050 [4,5]. Although osteoarthritis can affect multiple joints in the body, knee osteoarthritis (KOA) emerges as a predominant subtype of OA [6]. According to epidemiological data from 2019, out of the estimated 528 million individuals suffering from OA worldwide, approximately 365 million were diagnosed with KOA [7]. Clinically, it manifests primarily as knee pain that worsens after activity, accompanied by swelling, stiffness, and even joint deformity [8]. KOA is a leading cause of disability globally [9,10]. Furthermore, it significantly heightens the risk of various diseases, particularly cardiovascular conditions, and increases susceptibility to falls. This not only results in a decline in physical function, quality of life, and mental health for patients but also imposes a substantial economic burden on society [9,11]. The early diagnosis, timely identification, and intervention for KOA hold the potential to slow down, halt, or even reverse the disease, significantly enhancing patient prognosis and quality of life [12]. Simultaneously, these measures can mitigate the dependence on invasive and costly treatment modalities.

Currently, the diagnosis of osteoarthritis primarily relies on the patient's medical history, clinical symptoms, and medical imaging [13]. The first two methods offer limited insights into the joint's condition. While CT scans raise concerns due to their higher radiation doses, MRI procedures are relatively expensive. Radiographs (or plain films) serve as the primary diagnostic modality. These images allow for diagnosis based on indicators such as joint space narrowing, subchondral bone sclerosis, cystic changes, and osteophyte formation. Additionally, X-rays play a crucial in ruling out differential diagnoses such as stress fractures and avascular necrosis. A characteristic radiographic presentation includes reduced joint space, subchondral bone sclerosis, cystic changes, and the formation of lip-like or spiculated osteophytes along joint margins. In advanced stages, joint space may be completely obliterated, leading to joint surface deformities, potential subluxation, and malformation, although ankylosis remains rare. If fragments of cartilage or osteophytes detach, they can form loose bodies within the joint, colloquially referred to as "joint mice." If these bodies contain bony or calcified components, they become discernible on X-rays.

Presently, the widely accepted diagnostic criteria for knee osteoarthritis are the Kellgren–Lawrence (KL) grading system [14] and the atlas provided by the Osteoarthritis Research Society International (OARSI). However, both diagnostic approaches are susceptible to reader subjectivity. Due to subtle manifestations in the early stages and inconsistent grading, misdiagnoses are not uncommon. Hence, the development of a computer-assisted diagnostic tool is of paramount importance [15]. In recent years, the rapid development of artificial intelligence has found widespread applications in the intelligent recognition and auxiliary diagnosis of osteoarthritis. For instance, Lim et al. [16] proposed a method utilizing Deep Neural Network (DNN) and statistical data for early osteoarthritis recognition. This approach provides an early diagnostic tool for both physicians and patients, potentially alleviating physicians' workload in clinical settings. Similarly, Yeoh et al. [17] employed transfer learning to convert two-dimensional pre-trained weights into three-dimensional ones, aiding the precise prediction of knee arthritis using deep learning models. However, this method enables precise prediction under limited hardware conditions and lacks stability testing, such

as five-fold cross-validation. Nasser et al. [18] presented a convolutional neural network capable of identifying shape and texture to classify early knee osteoarthritis from X-ray images. While this offers an automatic identification approach, the accuracy rate stands at a mere 74.08%, leaving room for improvement. Additionally, Schiratti et al. [19] demonstrated the prospective application of deep learning in predicting osteoarthritis from MRI images, showing its potential in assisting physicians with diagnoses. Ren et al. [20] enhanced semantic information extraction by integrating features from ResNet and MobileNet with multi-scale feature fusion, preserving the high-resolution details provided by the lower layers of the network. This approach improved the model's ability to utilize multi-scale information in KOA, achieving an impressive accuracy of 84.88% in external tests. Despite these advancements, challenges persist in deep learning image recognition for osteoarthritis classification, including low recognition accuracy, oversimplified models, and the inability to fully maximize model performance. Additionally, most proposed models fail to address the critical textural and detailed information present in KOA, which necessitates deeper learning models to extract finer feature maps and richer semantic information. Moreover, as network depth increases, issues such as gradient vanishing can occur, hindering the model's effectiveness.

The objective of this study is to develop a high-accuracy algorithm for osteoarthritis detection, aiming to assist physicians in diagnosis, significantly reduce their workload, and increase the detection rate of osteoarthritis. To achieve this goal, we propose a model called OA-HybridCNN (OHC), a hybrid model based on DenseNet, Depthwise Separable Convolution, and Residual Blocks. Initially, CT images of osteoarthritis are input into DenseNet for preliminary feature extraction. This process efficiently extracts input features through repetitive use of features under certain parameter constraints. Subsequently, Residual Blocks are incorporated, adding a "shortcut" to directly connect input and output, effectively addressing the vanishing gradient problem and enhancing the model's expressiveness. The output of the Residual Blocks undergoes further feature extraction in the Depthwise Separable Convolution, enhancing the model's recognition capability. This method performs convolution operations separately on each channel, followed by a combination using 1x1 convolutions. This approach significantly reduces computational complexity and model parameter volume, minimizing interference between different channel inputs and effectively extracting features from the input. Finally, the results of feature extraction are input into the fully connected layer for classification. In predicting osteoarthritis, the OHC model has demonstrated superior performance with a high prediction accuracy rate. This achievement positions the model as a promising candidate for clinical application and serves as a valuable reference for physicians' diagnoses. The workflow of this study is illustrated in Fig 1.

## 2. Methods

### 2.1. Data sources

In this study, the Osteoarthritis Prediction database from Kaggle serves as the training set, and the Osteoarthritis_assignment database is the test set. Both databases are classified according to the Kellgren-Lawrence grading system, a widely

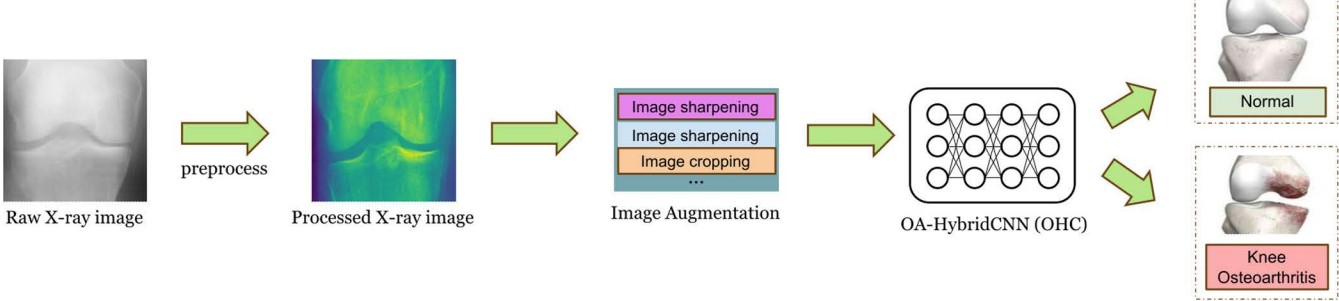

**Fig 1. The study workflow.**

used clinical methodology for assessing knee osteoarthritis severity [21]. Recognized by the World Health Organization as the standard grading criterion for OA research, this system categorizes knee joints into grades 0 (normal knee joint), I, II, III, and IV based on their radiographic appearance. However, this study focuses solely on KOA diagnosis. Therefore, we divided the dataset into two categories: 0 for normal and 1 for the presence of KOA. Fig 2 shows a visual representation of these two grades.

While the Kellgren-Lawrence (KL) grading system offers valuable insights, it lacks specifics on individual joint characteristics and lateral aspect changes. To address this gap, the Osteoarthritis Research Society International (OARSI) published a reference atlas for osteoarthritis staging [22]. This atlas refines grading based on features such as femoral osteophytes (FO), tibial osteophytes (TO), and joint space narrowing (JSN), providing a more contemporary radiographic grading approach tailored to osteoarthritis severity.

## 2.2. Ethics statement

All data are from public databases and do not involve ethical approval. This article does not contain any studies with human participants conducted by the authors. All data were obtained from public databases and did not involve patient consent.

## 2.3. Preprocessing data

To optimize our model's performance, we executed a series of preprocessing steps on the database. Initially, we randomized and shuffled the dataset, dividing it into training and test sets with an 8:2 ratio. Subsequently, we applied histogram equalization to the color space of the images and enhanced their contrast. These measures were taken to improve the model's comprehension and interpretive capabilities regarding the image data.

Moreover, due to the imbalance in the quantity of normal and osteoarthritis-affected samples within the dataset, we implemented a data augmentation strategy to enhance the model's recognition and understanding of anomalous images. Specifically, we subjected the training set images to operations such as rotation, cropping, color jittering, and noise injection. Subsequently, these processed images were reintegrated into the training set. These operations effectively mitigated the impact of data imbalance on the predictive performance of the model.

## 2.4. Model construction

### 2.4.1. Proposed fusion model.
The Depthwise Separable Convolution, first proposed by François Chollet in 2017 [23], was designed to reduce interference between neurons, decrease computational load, and minimize model size. In

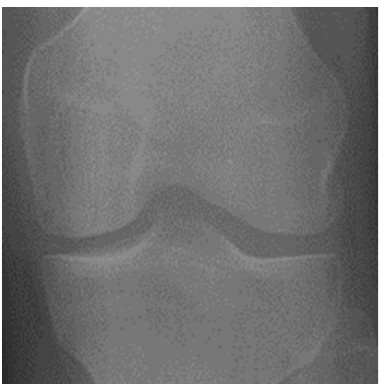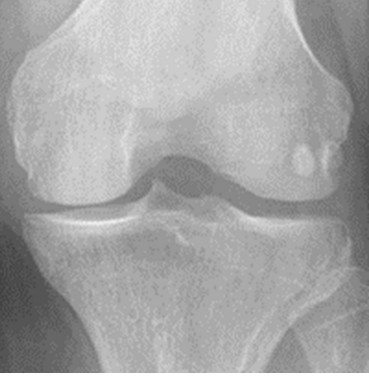

**Fig 2. Kellgren-Lawrence grade diagram (The left image depicts a normal knee joint's X-ray image, and the right one illustrates an X-ray from a patient with KOA).**

2015, Kaiming He et al. [24] introduced ResNet, incorporating residual blocks into the model. This innovative approach enabled the network to learn a residual function instead of a raw target function directly, effectively addressing vanishing and exploding gradient issues in deep learning. Additionally, DenseNet, introduced by Gao Huang et al. in 2016 [25], improved gradient flow and feature reusability by connecting each layer's feature map to all subsequent layers, optimizing parameter usage. While these models have garnered significant attention, their single network structure, whether overly simplistic or complex, often struggles to effectively learn input features, leading to decreased predictive performance. Consequently, developing hybrid models becomes imperative to optimize overall model performance.

In this study, we proposed the OHC model, a fusion model based on Residual Block, DenseNet, and Depthwise Separable Convolution. This model effectively utilizes Depthwise Separable Convolution and DenseNet for feature extraction, significantly reducing computational complexity and the number of model parameters, thereby enhancing computational efficiency. The inclusion of the Residual Block addresses the issue of gradient vanishing during training due to the model's depth. Ultimately, the fully connected layer classifies and outputs predictive probability values, classifying those above 50% as osteoarthritis. For the OHC model, we defined 2 DenseNet layers, 1 Depthwise Separable Convolution layer, 128 hidden neurons, 100 training rounds, a learning rate of 0.001, and a batch size of 64. These parameters effectively maximize model performance while preventing overfitting. Fig 3 shows the OHC model structure.

**2.4.2. Baseline models.** In this research, we performed comparative studies involving various foundational deep-learning models alongside our proposed OHC model. These models included ShuffleNet, NasNet, MobileNet, and EfficientNet. Furthermore, we constructed baseline models using DenseNet and ResNet, integral components of our OHC model, for comparative analysis. These models have been extensively employed and studied in the diagnosis of KOA, and many have served as baselines for state-of-the-art (SOTA) models proposed in recent research. Such widespread

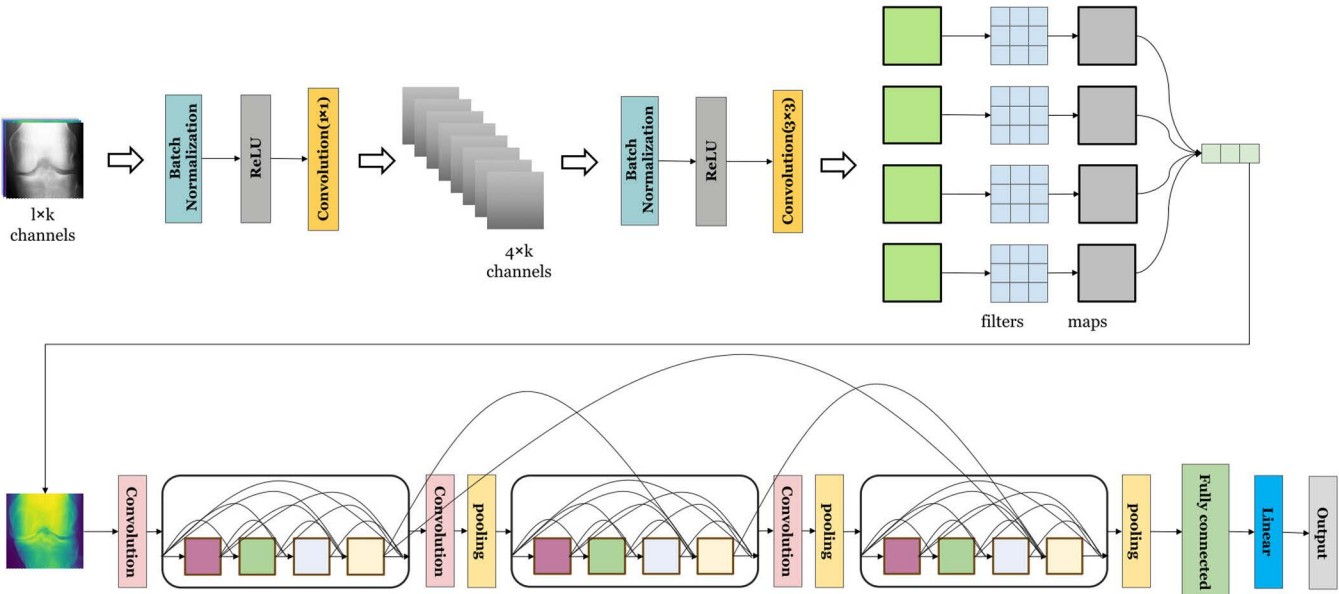

**Fig 3. Structure of the OHC model.** In this study, we employed Depthwise Separable Convolution to significantly reduce the model's parameters, decreasing complexity, thus expediting training and reducing the risk of overfitting. This technique enabled us to understand the images from various angles and scales, extracting a plethora of osteoarthritis features. Additionally, we incorporated Residual Blocks, effectively alleviating the vanishing and exploding gradient issues common in deep networks. This adaptation allowed for more profound network training. Leveraging pre-trained DenseNet, we extracted valuable information such as edge, color, texture, and other features from osteoarthritis images, significantly enhancing the efficiency and accuracy of our model. Moreover, we fused feature extraction from multiple layers, markedly improving the model's sensitivity compared to single models. This enhancement led to superior predictive accuracy.

validation underscores their effectiveness, thereby facilitating a rigorous assessment of our model's performance. By contrasting our OHC model with these established architectures, we aim to highlight its advantages, identify any shortcomings, and suggest directions for further enhancement.

ShuffleNet, a highly computationally efficient Convolutional Neural Network (CNN) architecture, was specifically designed for devices with limited computational power. It incorporates novel operations like Pointwise Group Convolution and Channel Shuffle, significantly reducing computational costs while maintaining precision. NasNet, a product of Neural Architecture Search (NAS), developed by researchers at Google Brain, automatically identifies the most effective model architecture. This approach outperforms manually designed networks and other exploration methods such as evolutionary algorithms or reinforcement learning, demonstrating exceptional performance in image recognition, surpassing that of ImageNet.

Like ShuffleNet, MobileNet is a lightweight model designed for mobile devices, optimizing computational speed and model size. This design allows it to maintain a relatively high degree of accuracy even when running on resource-limited devices. The key technology behind MobileNet is Depthwise Separable Convolution, which substantially reduces the model's complexity and size while preserving performance.

EfficientNet, on the other hand, is based on a novel concept called "compound scaling". Unlike traditional approaches, where researchers independently scale a network's depth (number of layers), width (number of channels per layer), or resolution (size of the input image) to improve performance, EfficientNet employs a more systematic approach. It considers the scaling of these three dimensions simultaneously, instead of adjusting them independently. This method has demonstrated exceptional performance in image recognition and has been widely applied in the field.

Previous research extensively validated and thoroughly explored these models, providing valuable references for establishing our model. However, our proposed model, being a novel architecture, requires improvement and validation in the context of established models.

## 2.5. Experiment set up

In this study, we utilized the Osteoarthritis Prediction database from Kaggle as the training set and the Osteoarthritis Assignment database as the test set. Specifically, the training set comprises 3,849 knee joint X-ray images, of which 1,602 images are labeled as healthy knee joints, and 2,247 images are derived from patients with osteoarthritis; the test set contains 3,615 images, including 1,589 healthy knee joint images and 2,026 osteoarthritis X-ray images. The volume of data is sufficient to ensure the effective convergence of the OHC model proposed in this paper.

For experimental validation, we utilized a 5-fold cross-validation and an external test to assess the model's performance. In the 5-fold cross-validation, the original training set was evenly divided into five subsets. We conducted five training and validation processes, using four subsets as training data each time and the remaining subset as validation data. This method resulted in five different combinations of training and validation sets, with each data point being used for validation once. The average of these five validation results served as the final performance evaluation of the model. This method avoids using fixed training and validation sets. Instead, the model was trained and validated multiple times on different subsets of data, enhancing its generalization and validating its stability. In the external test, we used all the pre-divided training sets as model training data and tested them with the designated test set. Compared to the 5-fold cross-validation, the external test allowed the model to fully utilize the entire training set, maximizing its performance under the given training conditions. Moreover, to optimize the parameters of the OHC model and all comparative models, we implemented gradient thresholding techniques. These adjustments included tuning the learning rate and batch size to ensure effective learning across samples while avoiding local optima and ensuring convergence speed. Additionally, we carefully managed the number of training epochs for each model to allow full convergence without overfitting. Furthermore, callback strategies were employed across all models, selecting the epoch with the highest validation accuracy as the final training output to ensure optimal model performance."

All experiments in this study were executed on the Windows 11 Professional Edition system, leveraging the Python language within the Python 3.10.9 framework. We utilized packages such as Pytorch 2.0.1 + cu117, Scikit-learn, Sklearn 0.0.post 1, scipy 1.10.0, and matplotlib to support the model architecture and validate results. Our hardware configuration included an Intel Core i7 10750H CPU (base frequency 2.6GHz, maximum turbo frequency 5GHz, six cores/twelve threads) and an NVIDIA GeForce GTX 1080Ti GPU (memory size 8GB, memory bus width 128bit).

## 2.6. Model evaluation

In this study, we utilized several key performance metrics to assess the effectiveness of our model. These metrics include Accuracy (ACC), Recall (Rec), Precision (PRE), F1-score (F1), and the Receiver Operating Characteristic curve (ROC). True Positives (TP), True Negatives (TN), False Positives (FP), and False Negatives (FN) were calculated using their respective standard formulas.

ACC quantifies the proportion of correct predictions made by our model and is computed as follows [26–30]:

$$Accuracy = \frac{TP + TN}{TP + TN + FP + FN} \tag{1}$$

REC (Sensitivity), measures the model's ability to correctly identify positive instances and is calculated using the formula:

$$Recall = \frac{TP}{TP + FN} \tag{2}$$

PRE is defined as the ratio of correctly identified positive instances and is calculated as follows:

$$Precision = \frac{TP}{TP + FP} \tag{3}$$

The F1 score, which represents the harmonic mean of precision and recall, provides a balanced assessment of these metrics and is calculated using the formula:

$$F1\ score = \frac{2 * (precision * recall)}{precision + recall} \tag{4}$$

The Receiver Operating Characteristic (ROC) curve illustrates the sensitivity and specificity of a binary classifier, serving as a vital tool for assessing classification performance. It plots the True Positive Rate (TPR) against the False Positive Rate (FPR). In theory, a superior classification performance is indicated when the ROC curve is closer to the upper left vertex of the ROC space. Moreover, the integral metric, the Area Under the Curve (AUC) of the ROC, is computed to quantify the model's overall efficacy, ranging between 0 and 1.

## 3. Results

### 3.1. The result of 5-fold cross-validation

In this study, we employed a five-fold cross-validation method to assess both the stability and performance of our model and compared it with five benchmark models. Following extensive experimental verification and testing, we set the number of training rounds for the OHC model to 300, the learning rate to 0.0001, and the batch size to 64. These parameters were chosen with careful consideration for training time and overfitting. Notably, our results demonstrated the model's robust convergence without any signs of overfitting. Fig 4 (a) shows a visual representation of the OHC model's training process.

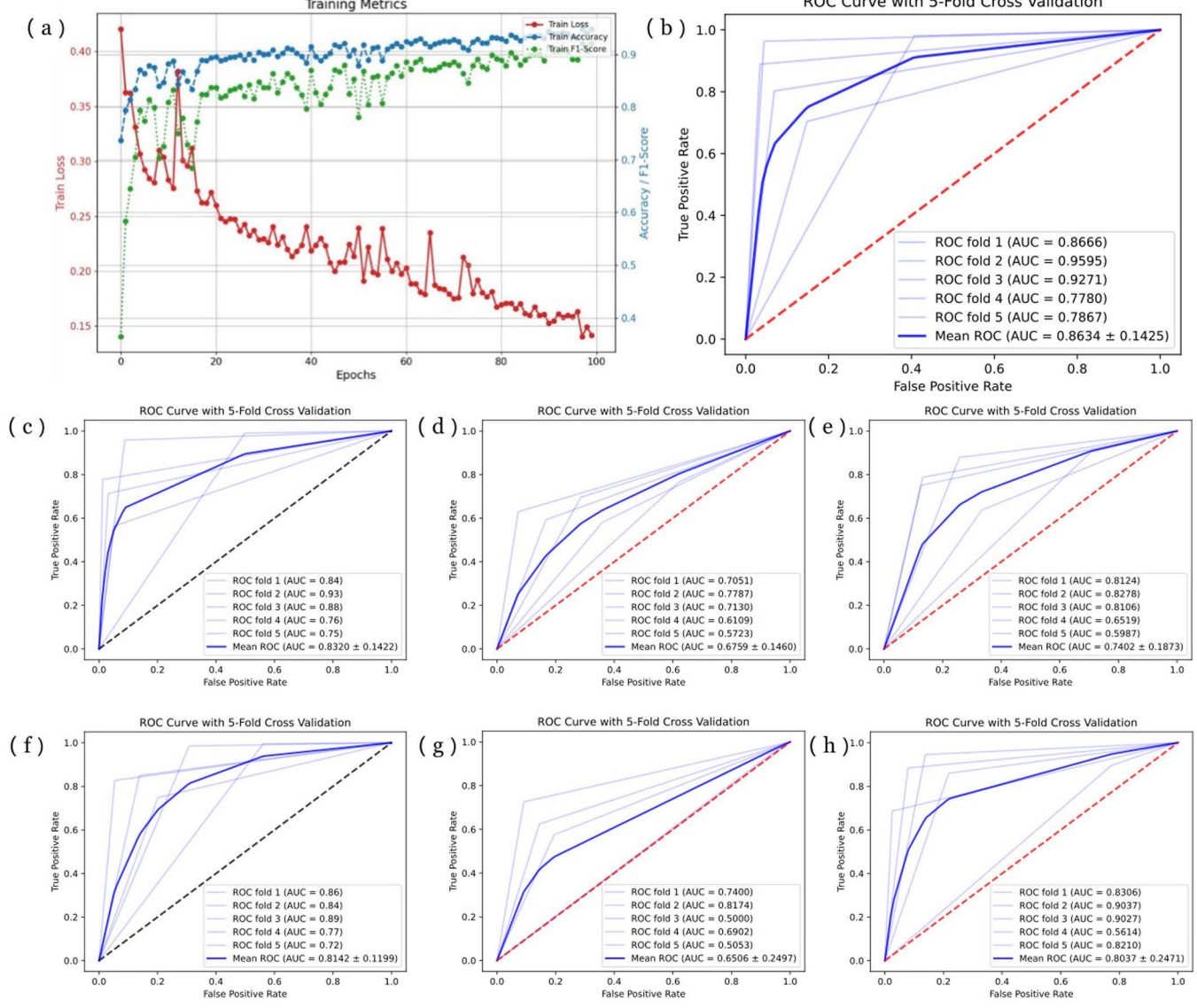

**Fig 4. Result of five-fold cross-validation. (a)** Training process of OHC model. **(b)** ROC curve of OHC model. **(c)-(h)** ROC curve of DenseNet, EfficientNet, MobileNet, NASNet, ResNet, and ShuffleNet, respectively (the black middle line represents the best two models).

We calculated the mean and 95% confidence intervals for ACC, PRE, REC, and F1 scores from the five-fold cross-validation results, to confirm the model's performance and stability. The ACC reached 87.42%±9.37%, PRE reached 88.51%±8.69%, REC reached 87.36%±10.11%, and F1 score reached 87.96%±9.160%. Notably, these metrics consistently demonstrated the narrowest 95% confidence intervals across all metrics when compared to all other models and the baseline, underscoring the exceptional performance and stability of our model. Table 1 presents a detailed comparison of the cross-validation results between the OHC model and baseline models. Additionally, we generated the ROC curve and the average ROC curve from the five-fold cross-validation, calculating the average AUC and its 95% confidence interval to be 86.34%±14.25%, signifying the model's robust performance. The ROC curve in the five-fold cross-validation of the OHC model is shown in Fig 4 (b).

**Table 1. Model performance in five-fold cross-validation.**

| | ACC | PRE | REC | AUC |
|---|---|---|---|---|
| **Our work** | **87.42%±9.37%** | **88.51%±8.69%** | **87.36%±10.11%** | **86.34%±14.25%** |
| DenseNet | 81.88%±12.40% | 84.84%±8.89% | 80.70%±13.89% | 80.70%±19.60% |
| ResNet | 81.00%±15.40% | 81.76%±14.87% | 80.39%±17.50% | 80.37%±24.71% |
| ShuffleNet | 85.23%±6.30% | 86.83%±5.88% | 85.09%±6.26% | 83.20%±14.22% |
| NasNet | 66.58%±13.29% | 61.20%±36.59% | 65.06%±17.68% | 65.06%±24.97% |
| MobileNet | 84.45%±10.02% | 86.11%±8.26% | 84.26%±11.20% | 81.42%±11.99% |
| EfficientNet | 74.66%±14.26% | 75.29%±11.00% | 74.03%±13.26% | 74.02%±18.73% |

Comparative analysis with the baseline models clearly emphasizes the advantages of our OHC model. As shown in Table 1, among the baseline models, the ShuffleNet model exhibited the highest effectiveness, with ACC, PRE, REC, F1, and AUC scores of 85.23%±6.30%, 86.83%±5.88%, 85.09%±6.26%, 85.95%±6.07%, and 83.20%±14.22%, respectively. While the stability of the ShuffleNet model marginally outperforms the OHC model, it is noteworthy that the mean values of all evaluation indicators were approximately 2% lower than those of the OHC model. ROC curves for the baseline models are presented in Fig 4(c-i). These comparisons unequivocally underscore the significant superiority of our proposed OHC model over the baseline models in predicting osteoarthritis.

## 3.2. Results of external validation

To optimize the OHC model's performance and leverage the full training set, we utilized the complete training for model training. Concurrently, the test set served as external validation samples to evaluate the model's performance. During external testing, we maintained the same model parameter settings as in the five-fold cross-validation: a learning rate of 0.0001, a batch size of 64, and 300 training rounds. The evaluation metrics employed were ACC, PRE, REC, and F1. In external testing, due to the predominance of KOA X-ray images over normal images in the training dataset, the OHC model tended to predict most samples as KOA, significantly skewing the model's judgment. To address this, we employed data augmentation techniques to balance the dataset and enhance the model's robustness. After implementing these adjustments, the OHC model exhibited remarkable performance with an ACC of 91.77%, PRE of 92.34%, REC of 91.36%, and F1 of 91.85%. These results highlighted the excellent performance and predictive accuracy of the model, promising its application in clinical auxiliary diagnosis. Table 2 presents a detailed comparison of the external testing results of the OHC model and the baseline models. As shown in Table 2, the parameter count for the OHC model is 5.3M, which is significantly reduced compared to the baseline models DenseNet and ResNet, and comparable to lightweight models such as ShuffleNet, EfficientNet, and NasNet-Mobile. Although it still has 2M more parameters than MobileNet, it shows an improvement of over 5% in predictive accuracy compared to MobileNet, effectively reducing parameter count while maintaining good predictive performance and enhancing computational efficiency. Additionally, we plotted the ROC curve and confusion matrix for the external testing, as shown in Fig 5(a) and (b), respectively. The AUC reached 91.18%, further validating the OHC model's exceptional predictive performance.

In this study, we applied the same method for external testing comparison with the baseline models. The ROC curves of these models are shown in Fig 5(c-i). Among the baseline models, ResNet demonstrated the best model performance, with ACC, PRE, REC, and F1 scores of 90.12%, 84.60%, 99.48%, and 91.44%, respectively. While its evaluation metrics were only approximately 1% lower than those of the OHC model, its excessive sensitivity resulted in a lower positive predictive rate and a significant number of false positives. This sensitivity flaw constitutes a critical issue in clinical testing. Therefore, after comprehensive consideration, we concluded that the OHC model exhibited the best predictive performance.

**Table 2. Model performance in external validation.**

|  | ACC | PRE | REC | F1 | AUC | Parameters (M) |
|---|---|---|---|---|---|---|
| **Our work** | **91.77%** | **92.34%** | **91.36%** | **91.85%** | **91.18%** | **5.4M** |
| DenseNet | 77.81% | 73.22% | 95.26% | 82.80% | 75.42% | 8.0M |
| ResNet | 91.12% | 84.60% | 99.48% | 91.44% | 92.02% | 25.6M |
| ShuffleNet | 88.11% | 96.02% | 82.18% | 88.56% | 88.92% | 5.4M |
| NasNet-mobile | 74.83% | 73.56% | 66.71% | 69.97% | 73.95% | 5.3M |
| MobileNet | 86.45% | 81.19% | 98.67% | 89.08% | 84.76% | 3.4M |
| EfficientNet | 87.94% | 90.77% | 87.36% | 89.03% | 88.02% | 5.3M |

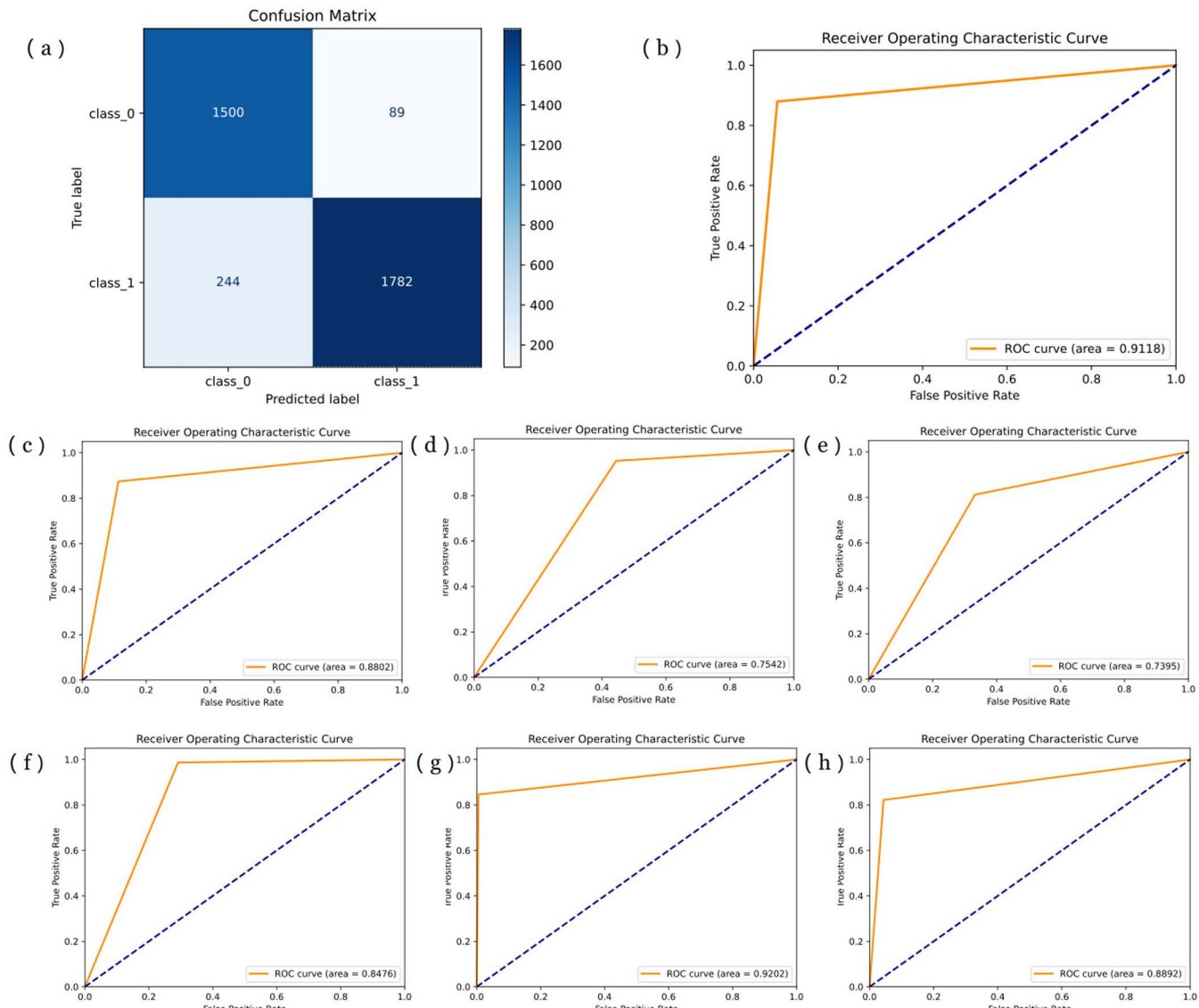

**Fig 5. Results of external validation. (a)** Confusion matrix of the OHC model. **(b)** ROC curve of OHC model. **(c)-(h)** ROC curve of DenseNet, Efficient-Net, MobileNet, NASNet, ResNet, and ShuffleNet, respectively.

## 3.3. Model visualization

To provide a more intuitive explanation of our proposed model's reasoning for diagnosing KOA and its focus on specific areas, we employed Gradient-weighted Class Activation Mapping (Grad-CAM) to produce visual representations of the model's activation heatmaps [31]. Fig 6 illustrates these heatmaps, where figure (a) depicts a normal knee joint and figure (b) shows a knee joint afflicted with KOA. In figure (a), the heatmap displays uniform activation across the joint, suggesting minimal anatomical abnormalities or stress, typical of a healthy knee. This uniformity indicates the absence of degenerative features characteristic of a normal joint. In contrast, figure (b) highlights concentrated areas of activation in regions critical to KOA pathology, such as the joint space and osteophytes. This indicates that the OHC model has accurately detected significant structural changes and abnormalities associated with KOA, including osteophyte formation, alterations in joint space, and cartilage wear. These heatmaps demonstrate that the OHC model effectively identifies and focuses on critical areas relevant to KOA, thus proving the effectiveness and enhancing the interpretability of our proposed model.

## 4. Discussion

This research aimed to provide clinicians with precise auxiliary diagnostic tools for osteoarthritis. To achieve this goal, we proposed the OHC model, a deep fusion model designed for accurate osteoarthritis prediction. The OHC model integrates the characteristics of Residual Block, DenseNet, and Depthwise Separable Convolution. DenseNet and Depthwise Separable Convolution effectively extract features from osteoarthritis images, while the Residual Block prevents gradient vanishing due to excessive model depth. The model's classification and output are processed through a fully connected layer. In the external validation, the OHC model demonstrated a predictive ACC of 91.77%, and an AUC of 91.18%, demonstrating excellent performance.

Compared to single models, the OHC model's strength lies in its integration of different models, capitalizing on the feature extraction capability of DenseNet, the computational efficiency and parameter optimization of Depthwise Separable Convolution, and the deep feature learning ability of Residual Block. These features enhance the model's performance in osteoarthritis image recognition tasks. Moreover, the OHC model is specifically optimized for osteoarthritis prediction tasks, leveraging data preprocessing techniques such as data augmentation and histogram equalization to improve

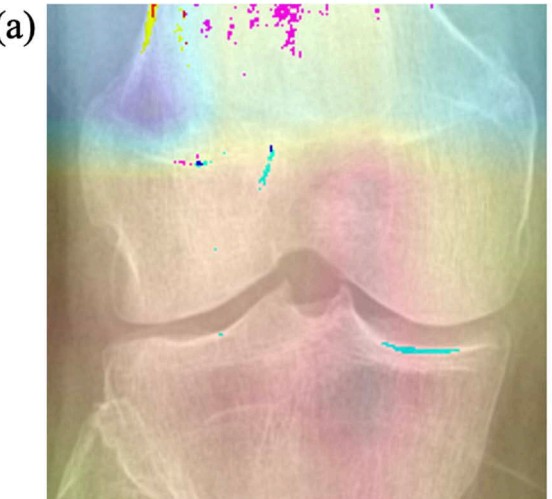 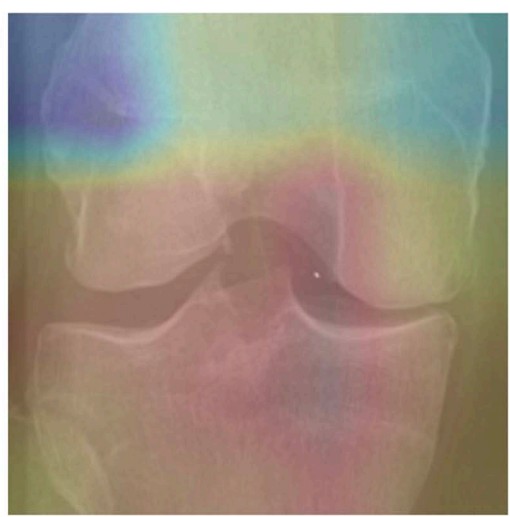

**Fig 6. The heat map of different categories (a) the heat map of healthy knee joint.** (b) the heat map of KOA patient's knee joint.

the model's recognition ability for inputs. These strategies lay the foundation for the OHC model's accurate osteoarthritis predictions.

OA is highly prevalent and can lead to severe complications. Early and accurate diagnosis is crucial for preventing OA progression and optimizing treatment strategies [32]. Currently, no specific laboratory tests for OA exist in clinical settings, Clinical diagnosis heavily relies on experience and subjective assessment, with X-rays being the most common imaging diagnostic tool [33]. Early osteoarthritis changes around the joint bones can be observed on X-rays [34].

The primary diagnostic methods for X-ray manual diagnosis of osteoarthritis include the Kellgren-Lawrence (K-L) grading criteria and the reference atlas published by the Osteoarthritis Research Society International (OARSI) [22]. Both methods provide semi-quantitative assessments, which are inherently subjective, leading to inconsistencies in diagnosis [35–37]. Physicians depend on subjective experience to interpret vast amounts of image data, making it challenging to avoid errors arising from the intrinsic limitations of human attention and perception [38].

The global shortage of medical imaging professionals, coupled with the extensive training and practice required for radiologists, results in an annual increase in the workload of medical imaging. This trend may become unsustainable in the future. In remote areas with insufficient medical personnel and lower diagnostic standards, achieving accurate diagnosis becomes even more challenging.

Studies analyzing bone microstructure in osteoarthritis [39] and the automatic diagnosis of knee osteoarthritis [40] date back more than three decades. AI can analyze historical medical image data, extracting texture features invisible to the naked eye [41]. This capability enhances knee OA diagnostic accuracy, improves physician efficiency, streamlines workflows, and reduces the occurrence of missed or incorrect diagnoses.

However, this study has limitations. It focused on binary classification, determining the presence or absence of osteoarthritis, without grading the severity of the condition. Standard X-ray examinations for Knee OA include full-length weight-bearing images of the lower extremities, anteroposterior and lateral views of the knee joint, and patellar axial views. However, in this study, we evaluated only the anteroposterior views of the knee joint, neglecting full-length weight-bearing images, lateral views, and patellar axial views.

Furthermore, this research focused solely on radiographic manifestations. In the early stages of OA, clinical imaging for early knee OA might yield negative results. Therefore, diagnosing early-stage OA in a clinical setting should consider other factors, including symptoms, physical examination findings, and laboratory test results.

In future studies, it would be beneficial to collaborate with medical institutions to acquire multi-angle X-ray images of the knee joint and collect more comprehensive patient data to optimize model and achieve testing on clinical datasets. This approach would facilitate a multimodal diagnosis of KOA, enhancing diagnostic accuracy not only for detecting KOA but also for grading its severity. These advancements could yield significant insights for treatment strategies and could be seamlessly integrated into clinical systems to provide physicians with an efficient decision-making tool. However, variations in clinical settings, such as geographic location, the extent of population aging, and ethnic distributions, may influence the model's efficacy. To address this, our deployment strategy includes an interface that allows physicians to annotate input data, which can then be used to further train the OHC model. This adaptive approach ensures that the model can be customized to reflect specific regional and demographic characteristics, thereby enhancing the precision of its outputs.

## 5. Conclusions

Knee osteoarthritis (KOA) poses a significant global burden, necessitating accurate and timely diagnosis to preserve patients' quality of life and prevent disease progression. To address these challenges, this study introduced an advanced deep learning model, the OA-HybridCNN (OHC), which integrates the strengths of ResNet and DensNet. This model stands out for its ability to circumvent the gradient vanishing issue observed in DensNet, demonstrating superior diagnostic accuracy. When compared to other models, the OHC consistently showed exceptional performance, achieving

remarkable results, including an accuracy of 91.77%, precision of 92.34%, and recall of 91.36% in external tests. Furthermore, in the five-fold cross-validation, the model exhibited an average AUC of 86.34% and ACC of 87.42%. These findings underscore the transformative potential of deep learning in X-ray image processing for KOA diagnoses. The proposed model holds the promise of seamless integration into clinical practices, alleviating diagnostic pressures on physicians, reducing the threshold for radiologists, expediting the KOA diagnostic process, and subsequently reducing patient waiting times. Future research could expand this model to multimodal diagnoses, achieving high-precision diagnostics through a synthesis of multi-dimensional information, such as multi-angle knee X-ray images and relevant patient data, aligning with the trend towards precision medicine.

## Supporting information

**S1 Data. Supporting information.**
(ZIP)

## Acknowledgments

The authors extend our sincere gratitude to the reviewers for their constructive suggestions for this article.

## Author contributions

**Conceptualization:** Yihan Liao, Guang Yang, yun lu.

**Data curation:** Wenjin Pan.

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
