## [Decision Letter · Decision Letter 0]

20 Jan 2025

PONE-D-24-51650OA-HybridCNN (OHC): An Advanced Deep Learning Fusion Model for Enhanced Diagnostic Accuracy in Knee Osteoarthritis ImagingPLOS ONE

Dear Dr. lu,

Thank you for submitting your manuscript to PLOS ONE. After careful consideration, we feel that it has merit but does not fully meet PLOS ONE’s publication criteria as it currently stands. Therefore, we invite you to submit a revised version of the manuscript that addresses the points raised during the review process.

We look forward to receiving your revised manuscript.

Kind regards,

Julfikar Haider

Academic Editor

PLOS ONE

Journal Requirements:

[Jiangsu Key Laboratory of Human and Animal Diseases, "Protecting Xin" Fund Project (HX2402), Kunshan City Science and Technology Special (Social Development) Guiding Project (KSZ2338).].

Additional Editor Comments:

See the comments from the reviewers

Reviewers' comments:

Reviewer's Responses to Questions

**Comments to the Author**

1. Is the manuscript technically sound, and do the data support the conclusions?

Reviewer #1: Yes

Reviewer #2: Partly

2. Has the statistical analysis been performed appropriately and rigorously? 

Reviewer #1: Yes

Reviewer #2: No

3. Have the authors made all data underlying the findings in their manuscript fully available?

Reviewer #1: No

Reviewer #2: Yes

4. Is the manuscript presented in an intelligible fashion and written in standard English?

Reviewer #1: Yes

Reviewer #2: Yes

5. Review Comments to the Author

Reviewer #1: The manuscript presents a novel deep learning model, OA-HybridCNN (OHC), designed to improve diagnostic accuracy in knee osteoarthritis imaging by integrating ResNet and DenseNet architectures. The study claims that the OHC model outperforms existing models in accuracy, precision, and recall, demonstrating its potential for clinical application. Overall, the manuscript is well-structured and provides a comprehensive analysis of the model's performance. However, there are areas that require further clarification and improvement to enhance the manuscript's clarity and impact.

Comments:

1. The introduction provides a solid background on knee osteoarthritis and the need for improved diagnostic tools. It clearly articulates the research problem and objectives. However, it would be beneficial to include more recent statistics or studies to support the claims about the prevalence and impact of knee osteoarthritis.

2. The literature review covers relevant studies on deep learning applications in osteoarthritis diagnosis. However, it could be more comprehensive by discussing recent advancements in similar models or alternative approaches. Additionally, integrating a critical analysis of existing methods would strengthen the rationale for developing the OHC model.

3. The methodology is detailed and generally appropriate for the study's objectives. However, some aspects lack clarity, such as the specific criteria for selecting baseline models for comparison. Additionally, more information on how hyperparameters were optimized would enhance reproducibility.

4. The results are presented clearly with appropriate use of tables and figures. However, there is a need for more detailed statistical analysis to support claims of superiority over baseline models. Including confidence intervals or significance testing would add rigor to the findings.

5. The discussion effectively relates the findings to the research questions and prior literature. It acknowledges some limitations but could benefit from a deeper exploration of potential biases or confounding factors in the study design. Additionally, discussing future research directions would enhance this section.

6. The conclusion succinctly summarizes key findings and contributions but lacks actionable recommendations for clinical practice or future research. Providing specific suggestions would increase its practical relevance.

7. The references are generally relevant and formatted correctly. However, there are instances where more recent sources could be included to reflect current advancements in the field.

Major revisions are required before considering acceptance for publication. Addressing the points mentioned above will improve clarity, depth, and overall quality of the manuscript, enhancing its contribution to the field of medical imaging diagnostics in osteoarthritis.

Reviewer #2: The manuscript titled "OA-HybridCNN (OHC): An Advanced Deep Learning Fusion Model for Enhanced Diagnostic Accuracy in Knee Osteoarthritis Imaging" introduced a hybrid deep learning model that integrates DenseNet and ResNet with Depthwise Separable Convolution. The study addressed a significant clinical challenge—automating the diagnosis of knee osteoarthritis (KOA)—and demonstrated promising results. However, some methodological and analytical gaps need to be addressed to ensure the robustness and impact of the proposed approach.

1- The study relies on two publicly available datasets (Kaggle and Osteoarthritis_assignment), which may not represent real-world variability in imaging conditions or patient populations.

Only anteroposterior X-ray views are considered, excluding lateral and axial views, which are often critical in clinical diagnosis. Authors should include additional datasets with varying imaging protocols and patient demographics to test the model's robustness and generalizability.

2- The model focuses solely on the presence or absence of KOA, ignoring the severity grading (e.g., Kellgren-Lawrence grades) that is crucial for clinical decision-making. Authors should extend the model to perform multi-class classification based on severity levels (e.g., Kellgren-Lawrence grades).

3-The robustness of the model across diverse imaging protocols, scanners, and patient populations is not tested, limiting its applicability to broader clinical settings. Integration visualization techniques (e.g., Grad-CAM) to identify which image regions contribute to the model’s decisions should be provided.

4-Although the model claims computational efficiency, the manuscript lacks detailed benchmarking against lightweight architectures such as MobileNet in resource-constrained environments. Comparing the model's computational performance with lightweight architectures like MobileNet or EfficientNet, particularly for deployment in resource-limited settings should be provided.

5-While augmentation is mentioned, the manuscript lacks details on how augmentation impacts performance, particularly on external validation datasets.

6-Discuss how the model can be integrated into clinical workflows, including potential challenges and how they might be addressed.

6. PLOS authors have the option to publish the peer review history of their article (what does this mean? ). If published, this will include your full peer review and any attached files.

**Do you want your identity to be public for this peer review?** For information about this choice, including consent withdrawal, please see our Privacy Policy .

Reviewer #1: **Yes: ** Mehrad Aria

Reviewer #2: No

---

## [Author Response · Author response to Decision Letter 1]

10 Mar 2025

Reviewer #1: The manuscript presents a novel deep learning model, OA-HybridCNN (OHC), designed to improve diagnostic accuracy in knee osteoarthritis imaging by integrating ResNet and DenseNet architectures. The study claims that the OHC model outperforms existing models in accuracy, precision, and recall, demonstrating its potential for clinical application. Overall, the manuscript is well-structured and provides a comprehensive analysis of the model's performance. However, there are areas that require further clarification and improvement to enhance the manuscript's clarity and impact.

Comments:

1. The introduction provides a solid background on knee osteoarthritis and the need for improved diagnostic tools. It clearly articulates the research problem and objectives. However, it would be beneficial to include more recent statistics or studies to support the claims about the prevalence and impact of knee osteoarthritis.

Response: Thank you for your constructive feedback. As suggested, we have updated the introduction section to include the most recent statistics regarding the prevalence of osteoarthritis (Page 3, Lines 5-10).

2. The literature review covers relevant studies on deep learning applications in osteoarthritis diagnosis. However, it could be more comprehensive by discussing recent advancements in similar models or alternative approaches. Additionally, integrating a critical analysis of existing methods would strengthen the rationale for developing the OHC model.

Response: Thank you for your questions. In response to your suggestions, we have expanded the discussion to include recent advancements in deep learning models used for osteoarthritis diagnosis. We have also critically analyzed these existing methods, highlighting their limitations and the gaps that our proposed OHC model aims to address (Page 4, Lines 21-29, and Page 5, Lines 1-2).

3. The methodology is detailed and generally appropriate for the study's objectives. However, some aspects lack clarity, such as the specific criteria for selecting baseline models for comparison. Additionally, more information on how hyperparameters were optimized would enhance reproducibility.

Response: Thank you for your constructive feedback on our manuscript. In response to your comments, we have made several clarifications in the methodology section. In Section 2.3.2, we have now included a detailed explanation of the criteria used for selecting the baseline and comparison models. Furthermore, in Section 2.4, we have expanded our discussion on the strategies for hyperparameter optimization. We have described the specific techniques and methods employed, along with the rationale behind each choice to ensure that our methodology is transparent and reproducible (Page 7, Lines 18-23, and Page 9, Lines 6-12).

4. The results are presented clearly with appropriate use of tables and figures. However, there is a need for more detailed statistical analysis to support claims of superiority over baseline models. Including confidence intervals or significance testing would add rigor to the findings.

Response: Thank you for your suggestion. To address this, we have included 95% confidence intervals for all evaluation metrics in our five-fold cross-validation, and we also added a more detailed description of the 95% confidence interval in the five-fold cross validation in section 3.1 (Page 10, Lines 23-28, and Tabel 1).

5. The discussion effectively relates the findings to the research questions and prior literature. It acknowledges some limitations but could benefit from a deeper exploration of potential biases or confounding factors in the study design. Additionally, discussing future research directions would enhance this section.

Response: Thank you for your suggestion. Based on your suggestions, we have added potential biases and confounding factors in the Discussion section that may affect the results of the study. We now provide a thorough analysis of these elements within the context of our study design, enhancing the transparency and reliability of our conclusions. Furthermore, we have enriched the section by outlining possible future research directions and improvements for our model (Page 14, Lines 18-29).

6. The conclusion succinctly summarizes key findings and contributions but lacks actionable recommendations for clinical practice or future research. Providing specific suggestions would increase its practical relevance.

Response: Thank you for your insightful suggestion. We have revised the conclusion to include specific actionable recommendations for the future research and clinical deployment of our model (Page 15, Lines 14-17).

7. The references are generally relevant and formatted correctly. However, there are instances where more recent sources could be included to reflect current advancements in the field.

Response: Thank you for this feedback. We have included more recent references to reflect the current advances in the field.

Reviewer #2: The manuscript titled "OA-HybridCNN (OHC): An Advanced Deep Learning Fusion Model for Enhanced Diagnostic Accuracy in Knee Osteoarthritis Imaging" introduced a hybrid deep learning model that integrates DenseNet and ResNet with Depthwise Separable Convolution. The study addressed a significant clinical challenge—automating the diagnosis of knee osteoarthritis (KOA)—and demonstrated promising results. However, some methodological and analytical gaps need to be addressed to ensure the robustness and impact of the proposed approach.

1- The study relies on two publicly available datasets (Kaggle and Osteoarthritis_assignment), which may not represent real-world variability in imaging conditions or patient populations.

Only anteroposterior X-ray views are considered, excluding lateral and axial views, which are often critical in clinical diagnosis. Authors should include additional datasets with varying imaging protocols and patient demographics to test the model's robustness and generalizability.

Response: Thank you for your valuable comments and suggestions regarding our study. We acknowledge the limitation of using only two datasets (Kaggle and Osteoarthritis_assignment) and the restriction to anteroposterior X-ray views in our current model. We understand the importance of representing real-world variability and the need for a comprehensive dataset that includes diverse imaging conditions and patient demographics. We are in the process of collaborating with hospitals to access a broader range of data that includes lateral and axial views, which are crucial for a more accurate clinical diagnosis. The inclusion of these additional datasets will undoubtedly enhance the robustness and generalizability of our model. These developments and future plans have been added to the discussion section of our manuscript to inform readers of our efforts to address these gaps (Page 14, Lines 19-29).

2- The model focuses solely on the presence or absence of KOA, ignoring the severity grading (e.g., Kellgren-Lawrence grades) that is crucial for clinical decision-making. Authors should extend the model to perform multi-class classification based on severity levels (e.g., Kellgren-Lawrence grades).

Response: Thank you for your insightful suggestion. Regarding the model's current focus on binary classification for the presence or absence of knee osteoarthritis (KOA), we recognize the clinical significance of grading the severity of the condition, such as through the Kellgren-Lawrence grades. To make our research more clinically relevant, we are extending our model to perform multi-class classification that will allow us to classify the severity levels of KOA. This will enable more nuanced clinical decision-making and provide more detailed insights for healthcare providers. We have outlined these changes and our future research directions in the discussion section as suggested (Page 14, Lines 18-29).

3-The robustness of the model across diverse imaging protocols, scanners, and patient populations is not tested, limiting its applicability to broader clinical settings. Integration visualization techniques (e.g., Grad-CAM) to identify which image regions contribute to the model’s decisions should be provided.

Response: Thank you for your valuable feedback. In response to your suggestions, we have integrated Grad-CAM visualization techniques to enhance the interpretability of our Osteoarthritis Health Classifier (OHC) model. In Section 3.3 of our manuscript, we now provide a detailed description along with Figure 6, which displays heatmaps of class activation for both healthy knee joints and those affected by knee osteoarthritis (KOA). These visualizations clearly illustrate which areas of the images are most influential in the model's decision-making process. While we acknowledge the current limitations in testing the model across varied imaging protocols, scanners, and patient demographics, we are planning future studies to address these aspects. This will include validating the model in a more diverse set of clinical environments to ensure its robustness and applicability in broader settings. Detailed descriptions of these validations are provided in the discussion section (Page 11, Lines 14-27, and Page 14, Lines 18-29 and Figure 6).

4-Although the model claims computational efficiency, the manuscript lacks detailed benchmarking against lightweight architectures such as MobileNet in resource-constrained environments. Comparing the model's computational performance with lightweight architectures like MobileNet or EfficientNet, particularly for deployment in resource-limited settings should be provided.

Response: Thank you for these professional suggestions. In response to your comments, we have included a new benchmarking analysis in Section 3.2 of our manuscript. This analysis compares the OHC model directly with lightweight architectures such as MobileNet and EfficientNet, specifically focusing on their performance in resource-constrained environments. In Table 2, we now present a detailed comparison of the parameter counts and computational loads of each model. This table is complemented by a discussion on how the OHC model not only maintains a lower parameter count but also achieves superior prediction accuracy compared to these lightweight architectures.

5-While augmentation is mentioned, the manuscript lacks details on how augmentation impacts performance, particularly on external validation datasets.

Response: Thank you for your valuable feedback on the description of data augmentation in our manuscript. To address your concern, we have expanded Section 3.2 to include a more detailed analysis of how data augmentation affects the performance of our model. We believe these detailed explanations will substantially strengthen the manuscript by showcasing the methodological rigor behind our augmentation strategy (Page 11, Lines 11-29 and Page 12, Lines 1-13).

6-Discuss how the model can be integrated into clinical workflows, including potential challenges and how they might be addressed.

Response: Thank you for your insightful comment. We appreciate the opportunity to further elaborate on the integration of our model into clinical workflows. In real-world clinical applications, several challenges may arise, including variations in patient demographics, population distributions, and healthcare infrastructure across different deployment regions. We have incorporated these considerations and the corresponding solution into the discussion section to emphasize the feasibility of real-world deployment. We appreciate the your suggestion, as it allows us to enhance the clarity and applicability of our study (Page 14, Lines 18-29).

---

## [Decision Letter · Decision Letter 1]

25 Mar 2025

OA-HybridCNN (OHC): An Advanced Deep Learning Fusion Model for Enhanced Diagnostic Accuracy in Knee Osteoarthritis Imaging

PONE-D-24-51650R1

Dear Dr. lu,

We’re pleased to inform you that your manuscript has been judged scientifically suitable for publication and will be formally accepted for publication once it meets all outstanding technical requirements.

Kind regards,

Julfikar Haider

Academic Editor

PLOS ONE

Additional Editor Comments (optional):

Reviewers' comments:

Reviewer's Responses to Questions

**Comments to the Author**

1. If the authors have adequately addressed your comments raised in a previous round of review and you feel that this manuscript is now acceptable for publication, you may indicate that here to bypass the “Comments to the Author” section, enter your conflict of interest statement in the “Confidential to Editor” section, and submit your "Accept" recommendation.

Reviewer #1: All comments have been addressed

Reviewer #2: All comments have been addressed

2. Is the manuscript technically sound, and do the data support the conclusions?

Reviewer #1: Yes

Reviewer #2: Yes

3. Has the statistical analysis been performed appropriately and rigorously? 

Reviewer #1: Yes

Reviewer #2: N/A

4. Have the authors made all data underlying the findings in their manuscript fully available?

Reviewer #1: Yes

Reviewer #2: Yes

5. Is the manuscript presented in an intelligible fashion and written in standard English?

Reviewer #1: Yes

Reviewer #2: Yes

6. Review Comments to the Author

Reviewer #1: The authors punctually responded to the reviewers' comments, adequately motivating any non-responses, and generally improving the quality of the manuscript. The manuscript is well-written, contains interesting information, and is suitable for publication in its present form.

Reviewer #2: The OHC model represents a meaningful advancement in deep learning for KOA imaging. The revisions adequately address reviewer concerns, and the study’s limitations are transparently acknowledged.

7. PLOS authors have the option to publish the peer review history of their article (what does this mean? ). If published, this will include your full peer review and any attached files.

**Do you want your identity to be public for this peer review?** For information about this choice, including consent withdrawal, please see our Privacy Policy .

Reviewer #1: **Yes: ** Mehrad Aria

Reviewer #2: No

---

## [Editor Report · Acceptance letter]

PONE-D-24-51650R1

PLOS ONE

Dear Dr. lu,

I'm pleased to inform you that your manuscript has been deemed suitable for publication in PLOS ONE. Congratulations! Your manuscript is now being handed over to our production team.

Kind regards,

on behalf of

Dr. Julfikar Haider

Academic Editor

PLOS ONE